# Global Analysis of RNA-Dependent RNA Polymerase-Dependent Small RNAs Reveals New Substrates and Functions for These Proteins and SGS3 in Arabidopsis

**DOI:** 10.3390/ncrna7020028

**Published:** 2021-04-27

**Authors:** Xia Hua, Nathan D. Berkowitz, Matthew R. Willmann, Xiang Yu, Eric Lyons, Brian D. Gregory

**Affiliations:** 1Department of Biology, University of Pennsylvania, Philadelphia, PA 19104, USA; huaxia@sas.upenn.edu (X.H.); nathan.berkowitz.gcb@gmail.com (N.D.B.); yuxiang1@sas.upenn.edu (X.Y.); 2Genomics and Computational Biology Graduate Group, Perelman School of Medicine, University of Pennsylvania, Philadelphia, PA 19104, USA; 3School of Integrative Plant Science, Cornell University, Ithaca, NY 14853, USA; mrw6@cornell.edu; 4School of Plant Sciences, University of Arizona, Tucson, AZ 85721, USA; ericlyons@email.arizona.edu; 5CyVerse, University of Arizona, Tucson, AZ 85721, USA

**Keywords:** RNA silencing, siRNA, posttranscriptional gene silencing, transcriptional gene silencing, RNA-dependent RNA polymerase, SGS3, RDM12, siRNA-target RNA interactions, RNA-mediated silencing

## Abstract

RNA silencing pathways control eukaryotic gene expression transcriptionally or posttranscriptionally in a sequence-specific manner. In RNA silencing, the production of double-stranded RNA (dsRNA) gives rise to various classes of 20–24 nucleotide (nt) small RNAs (smRNAs). In *Arabidopsis thaliana*, smRNAs are often derived from long dsRNA molecules synthesized by one of the six genomically encoded RNA-dependent RNA Polymerase (RDR) proteins. However, the full complement of the RDR-dependent smRNAs and functions that these proteins and their RNA-binding cofactors play in plant RNA silencing has not been fully uncovered. To address this gap, we performed a global genomic analysis of all six RDRs and two of their cofactors to find new substrates for RDRs and targets of the resulting RDR-derived siRNAs to uncover new functions for these proteins in plants. Based on these analyses, we identified substrates for the three RDRγ clade proteins (RDR3–5), which had not been well-characterized previously. We also identified new substrates for the other three RDRs (RDR1, RDR2, and RDR6) as well as the RDR2 cofactor RNA-directed DNA methylation 12 (RDM12) and the RDR6 cofactor suppressor of gene silencing 3 (SGS3). These findings revealed that the target substrates of SGS3 are not limited to those solely utilized by RDR6, but that this protein seems to be a more general cofactor for the RDR family of proteins. Additionally, we found that RDR6 and SGS3 are involved in the production of smRNAs that target transcripts related to abiotic stresses, including water deprivation, salt stress, and ABA response, and as expected the levels of these mRNAs are increased in *rdr6* and *sgs3* mutant plants. Correspondingly, plants that lack these proteins (*rdr6* and *sgs3* mutants) are hypersensitive to ABA treatment, tolerant to high levels of PEG8000, and have a higher survival rate under salt treatment in comparison to wild-type plants. In total, our analyses have provided an extremely data-rich resource for uncovering new functions of RDR-dependent RNA silencing in plants, while also revealing a previously unexplored link between the RDR6/SGS3-dependent pathway and plant abiotic stress responses.

## 1. Introduction

With 71% of the world’s population currently experiencing water scarcity [1] and the projected increase in human population to 12.3 billion people by 2100 [2], there is an urgent need to develop crops that can survive under stressful conditions to meet the future demand for food. In order to achieve this goal, a better understanding of the mechanisms that allow plants to survive in extreme conditions is required. The plant hormone abscisic acid (ABA) is a major regulator of plant response to abiotic stresses. Exogenous ABA treatment, a method universally applied to mimic abiotic stresses, causes stomata closure and induces the expression of stress-responsive genes [3,4]. Previous studies reported that small RNAs (smRNAs) can regulate the ABA signaling pathway and play critical roles in the stress tolerance of *Arabidopsis thaliana* (hereafter Arabidopsis) [5]. However, some of the proteins involved in the synthesis of these populations of smRNAs are still unknown.

In eukaryotes, smRNAs are characterized as noncoding RNAs that are usually 20–24 nucleotides (nt) long [6]. In plants, the two main classes of smRNAs are microRNAs (miRNAs) and short interfering RNAs (siRNAs), and are differentiated from one another based on their biogenesis, targets, and mechanisms of action [4]. The function of plant siRNAs is to posttranscriptionally regulate the abundance of transcripts encoding proteins involved in various processes. including the regulation of plant development, defense against biotic stresses, and response to abiotic conditions [7]. To date, three major subtypes of siRNAs have been identified in plant transcriptomes. These three classes are heterochromatic siRNAs (hc-siRNAs), trans-acting siRNAs (ta-siRNAs), and natural antisense siRNAs (nat-siRNAs). In eukaryotes, biogenesis of all classes of siRNAs begins with the formation of a dsRNA precursor, which is typically synthesized from a single-stranded RNA (ssRNA) template by an RDR [8,9]. The resulting dsRNAs are cleaved into siRNAs by a Dicer-like (DCL) protein, of which there are four (DCL1–4) in Arabidopsis. The produced siRNAs are then incorporated into an Argonaute (AGO) protein (10 in Arabidopsis), resulting in the formation of an RNA-induced silencing complex (RISC). The siRNA-bound RISC (siRISC) complex then targets coding or noncoding sites in RNA target molecules through complementary base-pairing interactions between the bound siRNA and the target transcript. This ultimately results in the silencing of these target loci through transcriptional gene silencing (TGS) and/or posttranscriptional gene silencing (PTGS) mechanisms [10,11]. 

RDRs are important RNA silencing components found in viruses, plants, and *Caenorhabditis elegans*, while most animals have lost their RDRs during evolution [12]. Arabidopsis contains six RDRs (named RDR1–6) that are classified into two different clades based on the amino acid motif of their catalytic domain. RDR1, RDR2, and RDR6 make up the RDRα clade that shares the C-terminal canonical DLDGD amino acid motif, while the RDRγ clade consists of RDR3–5 that have an atypical DFDGD motif [13]. Plant RDR proteins are involved in various processes, including pathogen defense, development, and abiotic stress responses [8]. For instance, RDR1 is involved in the production of virus-derived siRNAs and plays an important role in plant defense against viral infection [14,15]. RDR2 is involved in the biogenesis of 24 nt heterochromatic siRNAs (hsiRNAs) and is the only RDR protein that functions in the RNA-directed DNA methylation (RdDM) pathway that facilitates heterochromatin formation through DNA methylation and histone modifications [10]. Through this pathway, RDR2-produced hsiRNAs interfere with the development of female gametophytes in Arabidopsis [16]. During the production of hsiRNAs, RNA-directed DNA methylation 12 (RDM12), which has XS/coiled-coil domains, binds to dsRNA with 5′ overhangs and works as a cofactor for RDR2 [17]. No specific RNA silencing functions have yet been assigned to the RDR3–5proteins, even though they are evolutionarily conserved and are present in other plant genomes, including rice (*Oryza sativa*) and tomato (*Solanum lycopersicum*) [12,18]. RDR6 plays a major role in the production of trans-acting siRNAs (ta-siRNAs) and natural antisense siRNAs (nat-siRNAs) [11,19]. These siRNAs silence genes that participate in the processes of plant development, reproduction, and stress responses. The production of siRNAs by RDR6 occurs in conjunction with its cofactor SGS3, which has a similar protein structure and function to RDM12. 

RDR6 is the best-characterized plant RDR and has the most-known functions and widest range of substrates among these proteins. For instance, ta-siRNAs, which are usually 21–22 nt long, are produced exclusively by RDR6 and SGS3 from *trans-acting siRNA* (*TAS*) generating loci [11]. There are four known *TAS* loci in Arabidopsis, of which *TAS3* has been the most studied. The process of ta-siRNA biogenesis starts with the transcription of *TAS3* into primary *TAS3* (pri-*TAS3*) transcripts by RNA polymerase II. These pri-*TAS3* RNAs are then recognized by miR390 at two distinct sites, where the 3′ recognition site is cleaved by AGO7 RISC. RDR6, with the help of SGS3, binds to the 5′ fragment of the cleavage product and synthesizes a dsRNA that is further processed by DCL4 to generate 21 nt ta-siRNAs [20,21]. Some of the well-known targets of these 21 nt ta-siRNAs are from an auxin response factor (ARF) family of transcription factors (e.g., ARF3) and are involved in the leaf development and phase change in Arabidopsis [11,22]. 

While the RDR proteins are known to be conserved in plants, fungi, and *C. elegans*, SGS3 is a plant-specific protein. SGS3 has three defined protein domains: a zinc-binding domain, an XS domain that specifically binds to RNA, and a coiled-coil domain that mediates protein–protein interactions [23,24]. Although it has long been suggested that SGS3 functions to stabilize cleaved RNA fragments during dsRNA formation, the precise role of SGS3 in smRNA biogenesis is still not defined [25]. Previous studies showed the 3′ fragments of the cleaved pri-*TAS* RNA product are quickly degraded in *sgs3* mutants [26], and SGS3, RDR6, as well as AGO7 all colocalize in small interfering bodies [27]. In total, these findings suggest that SGS3 stabilizes and protects pri-*TAS* RNA cleavage products. A more recent study showed that the cleavage event is not required for ta-siRNA production, casting doubt on the necessity of SGS3 during this process. However, the function of ta-siRNA production is largely impaired from a noncleavable TAS3 construct in *sgs3* knockout mutants, indicating the role for SGS3 in this processing goes beyond stabilization/protection [25]. Furthermore, previous evidence showed that SGS3 is required for RDR1-dependent silencing of eceriferum 3 (CER3), a gene involved in wax biosynthesis [28], which suggests a role for SGS3 in facilitating RDR1 function. If SGS3 has substrates outside of those used by RDR6 they have not been comprehensively interrogated.

To better understand the roles of the six Arabidopsis RDRs in smRNA biosynthesis on a transcriptome-wide scale, we employed a global search for endogenous RDR substrates and targets of RDR-dependent smRNAs. Sequencing of smRNA (smRNA-seq) and total RNA (RNA-seq) was performed in mutants of the six RDR proteins (*rdr1–6*) as well as *sgs3* and *rdm12* mutants to identify genomic loci that require the various RDRs, SGS3, or RDM12 for subsequent smRNA processing or transcriptional/posttranscriptional regulation. A list of well-characterized substrates was also found in this study, serving as a validation for the quality of our findings. Furthermore, this study introduces an unexplored link between SGS3 and RDR6 in the regulation of plant abiotic stress response, as well as evidence that SGS3 shares substrates of other Arabidopsis RDR proteins.

## 2. Materials and Methods

### 2.1. Plant Material and Growth Conditions

*Arabidopsis thaliana* ecotype Columbia-0 (Col-0) was used as the wild-type control for all studies. The other genotypes used in these analyses were CS24285 (*rdr6**–11*) and CS24289 (*sgs3**–11*) and were previously described [11], all other mutant lines, including CS66077 (*rdr1**–1*), CS66076 (*rdr2**–1*), SALK_071908 (*rdr3**–1*), SALK_088276 (*rdr4**–1*), SALK_132667 (*rdr5**–1*), and SALK_152144 (*rdm12**–1*), were obtained from the Arabidopsis Biological Resource Center (ABRC, Ohio State University). All plants used in this study were grown in growth chambers under long day conditions (16 h light and 8 h dark photoperiods) at 22 °C. For sampling, unopened flower buds were extracted from 30-day-old soil-grown plants of all genetic backgrounds.

### 2.2. Library Preparation

Sequencing libraries were prepared for smRNA-seq and total RNA-seq as previously described [29,30]. For gel-mediated size selection of library inserts, small RNAs (15–50 nt) were selected using high-percentage (15%) denaturing acrylamide gels. Following a limited RNA fragmentation reaction, 140–250 nt RNA fragments were purified using acrylamide gels for total RNA samples that were purified from Arabidopsis 30-day-old unopened flower buds. The resulting sequencing libraries were sequenced on an Illumina HiSeq2000 using the 50 nt single-end sequencing protocol as per manufacturer’s instructions (Illumina Inc.; San Diego, CA, USA).

### 2.3. Sequencing Read Processing and Alignment

Some of the sequencing libraries were sequenced more than once to achieve greater sequencing depth. Multiple runs for each library were pooled by concatenating FASTQ files. For reads with small inserts, often the 3′ sequencing adapter was included in the resulting sequencing reads. These adapter sequences were trimmed using cutadapt [31]. Trimmed and untrimmed reads were separately mapped to the Arabidopsis genome (TAIR10) using TopHat [32,33], an aligner which takes into account splice junctions. Reads that could not be trimmed or mapped were discarded. Two mismatches per read and a maximum edit distance of two were allowed. One hundred alignments per read was allowed due to the repetitive nature of some RDR substrates.

### 2.4. Read Count Computation

The number of reads that mapped to Arabidopsis transcripts and genomic bins were counted separately. Reads aligning to each transcript were counted using HTSeq [34] in a strand-specific manner, separating the read counts that aligned to the sense and antisense sequences. Genomic bins were generated by partitioning the genome into unbiased 500 nt bins. The number of reads mapping to each genomic bin was counted using bedtools [35].

### 2.5. Differential Abundance Analysis and RDR Substrate Calling

Differential abundance analysis between Col-0 and each of the mutant genotypes was computed using the R package edgeR [36] for all library types. Features were considered to be putative RDR substrates if they had a decrease in smRNA levels of at least 33% with a false discovery rate (FDR) below 0.1. The significantly more abundant transcripts in RNA-seq with a *p*-value < 0.05 as calculated by edgeR [36] were considered as RNA substrates or the targets of their smRNAs and used as the input for the gene ontology (GO) analyses.

### 2.6. Size Classification of smRNAs

Trimmed smRNA-seq reads were used as a proxy for intact smRNAs. For each genotype, the set of putative RDR substrate genomic bins was intersected with the smRNAs in Col-0. These were classified by length and 5′ nucleotide.

### 2.7. Gene Ontology (GO) Analysis

For each of the genotypes, TAIR10 gene IDs of the transcripts were inputted into DAVID [37] or agriGO [38] and lists of all genes in each mutant with a count number equal or greater than one were used as reference backgrounds.

### 2.8. Target Prediction and Analysis

The set of all trimmed smRNA-seq reads before mapping to a reference genome was computed for total read count, differential abundance, and hit-calling analyses. Any smRNAs with a level decrease of at least 33% with an FDR below 0.05 were used to query psRNATarget [39] for target sites among the set of Arabidopsis transcripts, excluding miRNAs.

### 2.9. ABA treatment

Seeds were surface-sterilized using a 30% bleach and 0.1% Tween-20 solution for 10 min and washed three times with sterile water. These sterilized seeds were planted on Murashige and Skoog (MS) medium plates and incubated at 4 °C for 48 h, then transferred to a growth chamber at 22 °C under long day conditions (16 h light, 8 h dark) for two days. After two days, germinated seedlings were transferred to MS plates supplemented with 0, 0.5, 1, 5, or 10 μM ABA. Root length was measured after five days of growth in the absence or presence of ABA treatment.

### 2.10. NaCl Treatment

Seeds were sterilized as described above. The sterilized seeds for germination experiments were planted on MS plates supplemented with 0 or 200 mM NaCl and incubated at 4 °C for 48 h, then transferred to a growth chamber at 22 °C under long day conditions (16 h light, 8 h dark) for seven days before analysis. The sterilized seeds for the root elongation experiment were planted on MS plates and incubated at 4 °C for 48 h, then transferred to a growth chamber at 22 °C under long day conditions (16 h light, 8 h dark) for two days. After two days, germinated seedlings were transferred to MS plates supplemented with 0 or 200 mM NaCl. Root length was measured after five days of growth in the absence or presence of NaCl treatment.

### 2.11. PEG8000 Treatment

Seeds were sterilized as described above. One filter paper was soaked into one petri plate with 2 mL sterile water. Seeds were then planted on the filter paper, incubated at 4 °C for two days, followed by 22 °C in a growth chamber for two days. Germinated seedlings were incubated in 15 mL water or 20% PEG8000 solution for 2 days.

## 3. Results

### 3.1. Analysis of Putative Arabidopsis RDR Substrates

To identify the potential smRNA-producing substrates of the six Arabidopsis RDRs, we performed smRNA-seq on 15–50 nt RNAs from 30-day-old *Arabidopsis thaliana* ecotype Columbia-0 (hereafter Col-0) unopened flower bud tissue and plants containing null alleles for each *RDR* gene (*rdr1–6* mutants) as well as *rdm12* and *sgs3* mutants. The prepared smRNA-seq libraries were sequenced and provided 50–160 mln mapped reads per library. TopHat [32,33] was used to map trimmed smRNA reads to TAIR10 annotated transcripts, including genes and transposons, and the number of smRNA reads aligning to each transcript was counted using HTSeq [34]. To determine reproducibility, we used the counts per million mapped reads (CPM) of Arabidopsis transcripts from HTseq in the R package edgeR [36] and found that the biological replicates of smRNA-seq from each genotype are highly correlated (all R^2^ values > 0.9) (Appendix A), indicating the high quality and reproducibility of our sequencing libraries. Using the R package edgeR [36], we identified annotated transcripts and transposons where smRNA-seq read counts were significantly different for the knockout mutant plants compared to Col-0. In all eight mutants tested (*rdr1*–*6*, *sgs3*, and *rdm12*) we identified a number of transcripts that had a significant decrease (smRNA abundance down by 0.33, FDR < 0.05; Chi-square test) in smRNA read counts as compared to in Col-0 flower buds (Table 1). These loci with significant decreases in smRNA abundance were interpreted as putative smRNA-producing RDR substrates in Arabidopsis.

To further investigate the functional significance of each RDR and their cofactors, the number of putative substrates that are associated with each of these proteins was analyzed. Interestingly, we noticed that many of the putative RDR substrates had a significant loss of smRNAs being processed from both strands of the genome (including those not used by RNA Polymerase II (Pol II)) (Table 1), which serves as direct evidence that antisense transcripts produced by RDRs can be used as direct templates for smRNA biogenesis as well as the transcripts transcribed by Pol II. From our sequencing data, RDR2 was found to have the greatest number of putative substrates among the six RDRs in Arabidopsis, including 12,488 substrates on the sense strand and 2302 substrates on the antisense strand. Among all RDR2 substrates, nearly 76% (9536 substrates) of these RNAs are annotated as transposon or transposable elements (Appendix A). This finding is consistent with the fact that hsiRNAs, which are produced from heterochromatin and transposable elements by RDR2, are the most abundant smRNA population in the cell [10]. To characterize what processes are regulated by RDR2, we performed a Gene Ontology (GO) analysis using DAVID [37] on transcripts that were interpreted as putative RDR2 substrates. We found that these substrates were involved in the regulation of the ubiquitination pathway, plant biotic defense, and sugar metabolic process (Appendix A). As a cofactor of RDR2, RDM12 should process similar substrates for smRNA biogenesis. Indeed, RDM12 shares 96.8% of its substrates (240 total substrates) with RDR2, while also leaving 98% of RDR2 putative substrates (12,275 substrates) that are not shared by RDM12, indicating RDR2 has functions in the absence of this cofactor protein.

We also searched for putative substrates that can be processed by RDR1. The identified substrates suggested that RDR1 may be involved in abiotic stress response as the smRNAs processed from *HOS10* (*AT1G35515*), a gene whose protein product is involved in responses to various abiotic stresses, and a gene from the SAUR-like auxin-responsive protein family named *SAUR43* (*AT5G42410*) were lost in *rdr1* mutants (Appendix A) [40,41]. We next investigated the putative substrates of the three γ RDRs, RDR3–5. We identified a single putative RDR3 substrate (*AT1TE61180*), and this one RNA also overlapped with putative substrates of both RDR4 and RDR5 (Table 2). Interestingly, RDR4 is involved in processing putative substrates, such as *MLP31* (*AT1G70840*), *XAL1* (*AT1G71692*), and *ABH1* (*AT2G13540*), which affect the protein abundance increase under SA treatment, plant phase transition, and ABA signaling, respectively [42,43,44]. The finding that smRNAs can be processed from the *ABH1* transcript provides a new and additional link between PTGS and the important plant hormone ABA. Additionally, we uncovered putative RDR5 substrates, such as *PEN3* (*AT1G59870*), which is involved in the response to biotic stresses [45]. These findings suggest that RDR4 and RDR5 may be involved in processes such as the response to biotic and/or abiotic stresses. In addition, RDR4 and RDR5 share 26 putative substrates that fall into both annotated genes and transposable elements, suggesting that they may share some similar functions and/or function somewhat redundantly (Table 2). In total, although there is some overlap in substrates, the unique substrates of the γ clade RDRs suggests they have functional significance to plant PTGS.

We next studied putative substrates that are dependent on RDR6 and/or SGS3 for their processing into smRNAs. We identified most of the previously discovered substrates in this analysis, including *auxin response factors* (*ARFs*), *auxin signaling F-boxes* (*AFBs*), *pentatricopeptide repeat* (*PPR*) superfamily transcripts, and *tetratricopeptide* (*TPR*) superfamily mRNAs [11,46,47] (Appendix A). Furthermore, most of the *TAS* loci showed significantly lower abundance of smRNAs in both *rdr6* and *sgs3* mutants compared with Col-0, consistent with the role of RDR6 and SGS3 in ta-siRNA biogenesis. In total, these results provided internal validation of our smRNA-seq library quality and the ability of our approach to identify bona fide RDR substrates. In addition, this analysis revealed that RDR6 and SGS3 share other putative substrates, such as *HAIRY MERISTEM1* (*HAM1*), *HAM3* and *TCP* transcription factor genes (*TCP2–4*, *TCP10*) (Appendix A), which are involved in plant development, suggesting new regulatory roles for RDR6 and SGS3 in this process [48,49,50]. Intriguingly, we found that 88% of the total SGS3 substrates do not overlap with those of RDR6, suggesting that SGS3 likely has independent functions, such as facilitating other RDRs, including those from the γ clade (Table 2). In fact, we found overlap of SGS3-dependent substrates with those of RDR1, RDR2, RDR4, and RDR5 in addition to those shared with RDR6. We also found a large number (78.8%) of SGS3-dependet 24 nt smRNAs are also dependent on RDR2 for their processing. Thus, SGS3 is likely an RNA binding cofactor for most of the Arabidopsis RDRs, but not required for the processing of all putative substrates for each RDR. More focus should be given to define RDR substrates that do and do not require SGS3 in the future.

### 3.2. Small RNA Breakdown Associated with Each Arabidopsis RDR

To investigate the smRNA populations associated with each of the RDRs and the two cofactors that were profiled (SGS3 and RDM12), we divided the Arabidopsis genome into unbiased 500 nt genomic bins, and smRNA-seq reads from the biological replicates of Col-0 were mapped to these genomic bins. These mapped smRNAs in Col-0 were used as a reference to identify the smRNAs that demonstrated a decrease in *rdr1–6*, *rdm12*, or *sgs3* mutant plants. To achieve this, we counted smRNA enrichment or loss in each of the genotypes based on smRNA length and 5′ nucleotide (Figure 1). Previous work has demonstrated that the 24 nt smRNAs with a 5′ adenosine bias are the most abundant smRNA class in Col-0, whereas 21 nt smRNAs tend to have a significant uridine bias as the terminal 5′ nucleotide in this genotype [51]. We focused first on the smRNAs that were lost in the absence of RDR6 function (*rdr6* mutant). As expected, we found that a large proportion of 21–22 nt smRNAs with 5′ adenosine bias was lost in the *rdr6* mutant, indicating RDR6 plays a crucial role in the biogenesis of this smRNA population. We also found that the smRNA populations that are decreased in *sgs3* mutant plants look very similar to the total population of Col-0 smRNAs, where 24 nt smRNAs are most affected followed by those that are 21 nt in length. This pattern of less abundant smRNAs is very distinct from that observed in *rdr6* mutant plants, providing further support for the hypothesis that SGS3 can act as a cofactor for other RDR proteins, including RDR2. In fact, there were 1028 smRNAs that demonstrated a significant decrease in their abundance in the *sgs3* mutant as compared to a total of 608 smRNAs that were significantly decreased in the *rdr6* mutants. Relatedly, among all the smRNAs lost in *sgs3* mutants, 59.7% of them were not significantly decreased in the absence of RDR6 function (*rdr6* mutant). These data suggest that the biogenesis of a larger population of smRNAs is dependent on SGS3 but not RDR6, and SGS3 may play roles beyond just facilitating RDR6-dependent smRNA processing. The overall composition of smRNAs that require the other RDRs (RDR1 and RDR3–5) and RDM12 for their processing are very similar to the population of smRNAs in wild-type plants, where 24 nt smRNAs are the most prominent with some 21–23 nt smRNAs also requiring these proteins for their biogenesis (Figure 1). Overall, these findings reveal that each RDR protein affects a subpopulation of the overall population of RDR-dependent smRNAs with RDR2 having the largest overall effect due to its function in the biogenesis of hsiRNAs in plant transcriptomes.

### 3.3. Total RNA-seq Data Revealed New Putative Functions for RDRs

Gene silencing in Arabidopsis ultimately results in the produced siRNAs binding to mRNA targets through complementary base-pairing interactions and subsequent TGS- or PTGS-directed gene silencing [10,11]. Thus, one would predict that RDR-derived smRNAs are not synthesized in an *rdr* knockout mutant, and as a consequence cause an increase in the abundance of the initial RDR substrate transcripts and an increase of the RDR-dependent siRNA target transcripts when compared with Col-0. Therefore, we used this logic to identify the putative substrate transcripts of RDRs and target transcripts of RDR-derived siRNAs. Specifically, we performed total RNA-seq on 30-day-old unopened flower bud tissue for Col-0, *rdr1–6*, *rdm12*, and *sgs3*. The prepared RNA-seq libraries were sequenced and provided 11–31 million mapped reads per library. TopHat [32,33] was used to map identified transcripts to TAIR10 annotated mRNAs, and the number of reads aligning to each transcript was counted using HTSeq [34]. To determine reproducibility, we used the counts per million mapped reads (CPM) of Arabidopsis transcripts from HTseq in the R package edgeR [36] and found that biological replicates of total RNA-seq from each genotype were highly (all R^2^ values > 0.87) correlated (Appendix A), indicating the high quality and reproducibility of our sequencing libraries. We then used edgeR to identify transcripts that displayed significantly higher levels (*p*-value < 0.05) in the mutants relative to Col-0. These collections of differentially abundant transcripts were then used in a GO analysis using the online tool DAVID [37]. The GO term results were divided into three categories: biological processes, cell components, and molecular functions. The GO terms that were significantly enriched (*p*-value < 0.05) and are shared by more than two genotypes were then further interrogated (Figure 2). In the GO terms specific to biological processes (Figure 2A), we found that *rdr1, rdr2, rdr4*, and *rdr5* had similar enriched GO terms, including photosynthesis, response to light, and abiotic stresses, suggesting similar regulatory roles of RDR1, RDR2, RDR4, and RDR5 proteins in these processes. RDR3 may play a regulatory function during plant response to heat conditions since *rdr3* mutant plants had seven significantly more abundant heat-related transcripts when compared to Col-0. Interestingly, the significantly more abundant transcript lists for *rdr6* and *sgs3* mutant plants share many abiotic stress GO terms, including response to water deprivation, salt stress, abscisic acid (ABA), and osmotic stress. These results suggest that the collaboration between RDR6 and SGS3 may have widespread significance in the regulation of plant abiotic stress responses. Conversely, we also found GO terms that were specific to *sgs3* and not *rdr6* mutant plants, which included response to cold and heat. Transcripts that represented these GO terms included *temperature-induced lipocalin* (*AT5G58070*), which is involved in thermotolerance, and a gene in heat shock gene family named *heat shock cognate protein 70-1* (*AT5G02500*) that is involved in plant heat tolerance as well as virus infection defense [52,53,54]. These data provide additional support for our findings that SGS3 has functions independent of RDR6 in various plant abiotic stress responses, and also works with other RDRs to regulate various processes in plants. In support of this latter hypothesis and previous findings [28], our GO analysis indicated that SGS3 and RDR1 both regulate a set of transcripts that are involved in the response to plant wounding, suggesting that SGS3 can also act as a cofactor for RDR1 to regulate various processes in plants, including the response to wounding.

To further study the function of RDRs and their cofactors in plant biology, we also studied the over-represented GO terms for cellular components and molecular functions amongst the populations of transcripts that showed significantly increased abundance in the six *rdrs* and two cofactor mutants. In general, the pattern of GO enrichments amongst the transcripts misregulated in the *rdr1*, *rdr2*, *rdr4*, *rdr5*, and *sgs3* mutants are similar for both cellular component (Figure 2B) and molecular function terms (Figure 2C), suggesting potentially similar regulatory functions among these four RDR proteins and SGS3 with regard to these specific cellular locales, processes, and functions. Specifically, *rdr1*, *rdr2*, *rdr4*, *rdr5*, and *sgs3* all have significant increases in abundance for transcripts enriched in cellular components, such as chloroplast and processes related to photosynthesis, suggesting the photosynthetic pathway might be affected by the loss of these proteins. Our data also showed that a lower number of GO terms are associated with *rdr3* mutants, which is consistent with its limited effect on smRNA populations in Arabidopsis plants. We also compared the GO term patterns for RDR2 and RDR6 with their respective cofactors. In the case of RDR2, RDM12 works as its cofactor [17], and as expected we found that *rdr2* and *rdm12* overlapped in many GO terms. However, *rdr2* mutants displayed enrichment for differentially abundant transcripts in many more terms as compared to *rdm12* mutants. This finding was not surprising given that 399 transcripts were differentially abundant in *rdm12* mutants; there were many more transcripts (559 total) significantly more abundant in *rdr2* mutants as compared to Col-0. These results provide further support for our findings that RDM12 acts as a cofactor to RDR2 at specific substrates, but RDR2 also has a significant level of substrates that do not require this RNA-binding protein (RBP). 

Previous findings demonstrated that SGS3 functions in RDR6-dependent smRNA biogenesis, especially in the processing of ta-siRNA, and this was further supported by our findings that many of the misregulated transcript GO terms were shared by these two proteins. However, we also found that there were a number of additional GO terms enriched for *sgs3* that were not shared with *rdr6* mutant plants, and this trend was true throughout all three GO term categories that were analyzed (Figure 2A–C). In fact, the RNA-seq experiments revealed that *rdr6* mutants had a total of 643 transcripts while *sgs3* mutants had 688 transcripts that were significantly more abundant in comparison with Col-0, and of these totals only one third of these transcript populations (193 transcripts) were common between the two mutant genotypes. Thus, our findings reveal that SGS3 has other functions beyond solely facilitating RDR6 in the processing of smRNAs that are dependent on this RDR for their biogenesis. 

### 3.4. The rdr6 and sgs3 Mutant Plants were Hypersensitive to ABA and Less Sensitive to Salt and PEG8000 Treatment

Through the careful analysis of our RNA-seq data, we found that transcripts related to abiotic stresses, including water deprivation, salt stress, and ABA response, were significantly more abundant in both *rdr6* and *sgs3* mutants as compared to Col-0 plants (Figure 2A). Thus, these processes appear to require both proteins for their proper regulation in plants. To test this possibility, we studied the responses of *rdr6* and *sgs3* mutant plants under these abiotic stress conditions/treatments. To observe whether *rdr6* and *sgs3* mutants have phenotypes under ABA treatment, two-day-old seedlings of Col-0, *rdr6*, and *sgs3* were treated with increasing concentrations of ABA (0, 0.5, 1, or 5 μM) on growth medium-containing plates for seven days, and root elongation was evaluated as a measure of seedling stress response. After seven days of growth in the presence or absence of ABA, the root length of both *sgs3* and *rdr6* seedlings was significantly (*p*-value < 0.05; Mann–Whitney test) shorter than Col-0 seedlings when the plants were treated with various concentrations of ABA as compared to when all genotypes were not subjected to this hormone treatment (Figure 3A,B). These data reveal that both *rdr6* and *sgs3* are hypersensitive to ABA treatment and unveil a new connection between these smRNA-producing proteins and response to this important plant hormone. 

ABA mediates the plant response to various abiotic stressors, including salt and water deprivation [55], which were response terms that were identified in our GO analysis of transcripts misregulated in the absence of both SGS3 and RDR6 function. To study whether *rdr6* and *sgs3* mutant plants are also required for proper response to these abiotic stresses with inappropriate responses consistent with their ABA hypersensitivity, seedlings of Col-0 and both mutants were planted either in petri plates containing 20% polyethylene glycol 8000 (PEG8000) solution or growth medium containing 200 mM NaCl (salt). When using 20% PEG8000 as a mimic for osmotic stress, because it is known to lower water potential in cells [56], we found that the two cotyledons of *rdr6* and *sgs3* mutant seedlings were able to fully open in the presence of this compound, while the majority of Col-0 seedling cotyledons remained unopened at day 4 (Figure 4A,B). This finding reveals that *rdr6* and *sgs3* mutant seedlings are more tolerant to PEG8000 treatment, suggesting that they are more resistant to osmotic stress than wild-type plants. We also treated two-day-old Col-0, *sgs3*, and *rdr6* seedlings with 200 mM NaCl for five days on growth medium-containing plates to observe the effect of salt stress on these three genotypes. We found that nearly all Col-0 seedlings displayed white cotyledons that lacked chloroplast development after five days of salt treatment, while more than 60% of *rdr6* or *sgs3* mutant plants developed normal green cotyledons that displayed chloroplast development even at this high salt concentration (Figure 4C–E), indicating the mutant seedlings are more salt tolerant than Col-0. In total, these results revealed that the smRNA-producing proteins SGS3 and RDR6 are hypersensitive to ABA and more resistant to corresponding abiotic stressors, such as salt and osmotic stress, as compared to Col-0. Thus, our findings suggest that an smRNA-mediated PTGS pathway is likely involved in the proper plant response to numerous abiotic stresses as well as proper response to the important plant hormone ABA.

To elucidate the potential siRNA-mediated silencing targets that result in the ABA hypersensitivity of *rdr6* and *sgs3* mutant seedlings, we further analyzed our smRNA-seq and total RNA-seq libraries to identify the target RNAs of those siRNAs that depend on both RDR6 and SGS3 for their biogenesis. Plant siRNAs bind to their targets through complementary base-pairing interactions along nearly their full length, making it easier to predict potential target RNAs of plant siRNAs by searching for sites with near-perfect complementarity in silico. In order to predict targets for SGS3- and RDR6-dependent siRNAs, smRNA populations significantly lost (*p*-value < 0.01; EdgeR) in both the *rdr6* and *sgs3* mutants were inputted into the target search tool psRNATarget [39] (Appendix A). Next, target RNAs identified for this class of endogenous siRNAs by psRNATarget were overlapped with transcripts that are significantly more abundant (*p*-value < 0.05; EdgeR) in *rdr6* and *sgs3* mutants than Col-0 as identified from our RNA-seq analyses. Using this approach, we identified 86 and 100 putative RDR6- and SGS3-dependent smRNAs, respectively, and their predicted target transcripts that are significantly more abundant in *rdr6* and *sgs3* mutants as compared to Col-0 plants. Among the 16 predicted targets shared by RDR6 and SGS3, two of them are quite intriguing in that they have strong connections to proper ABA responses in plants. Specifically, we identified *nine-cis-epoxycarotenoid 3* (*NCED3*), which catalyzes the first step in abscisic-acid biosynthesis [57,58], and *MAP kinase kinase 9* (*MKK9*), which has been identified for its importance to abiotic stress response [59,60], as potential target RNAs of RDR6- and SGS3-dependent smRNAs. This analysis suggests a possible mechanism whereby lack of RDR6 or SGS3 in the knockout mutants fails to produce siRNAs that target *NCED3* and *MKK9*, and as a consequence, cause the hypersensitivity to ABA, resulting in tolerance to osmotic and salt stresses as observed in the *rdr6* and *sgs3* mutant seedlings. This hypothesis will require further testing in the future.

## 4. Discussion

Our study conducted the first global transcriptome analysis of all RDRs and their two known cofactors in Arabidopsis. We provide insights into new substrates processed by RDRs resulting in smRNA biogenesis, new smRNA populations produced by RDRs, and global determination of targets of RDR-derived siRNAs. A genome-wide search for putative substrates in RDR-dependent smRNA biogenesis validated previously discovered substrates while also uncovering novel RDR substrate populations. Plant siRNAs bind to their targets through complementary base-pairing interactions throughout most or all of their full length, often resulting in siRNAs to feedback target their own substrates. Thus, a functional analysis of putative RDR substrates will help to understand the roles of RDRs in plant biology. Our findings suggest uncharacterized functions for all interrogated RDRs and their cofactors. For instance, our findings suggest a role for RDR1 in abiotic stress response while revealing that RDR2 and RDM12 may be involved in ubiquitination, plant biotic defense, and sugar metabolic processes. Additionally, we found that two of the RDRγ clade proteins (RDR4 and RDR5) may share similar functions in processes such as responses to biotic or abiotic stresses, while RDR6 and SGS3 may utilize *HAM* and *TCP* transcripts as their substrates to regulate shoot, root, or leaf development.

RNA interference is the process of siRNAs binding to their target RNAs through complementary base-paring interactions followed by the silencing of those target transcripts, usually through their degradation. Therefore, a loss of RDR-dependent smRNAs in *rdr* or their cofactor mutant plants will result in the accumulation of their target RNAs. Our global search for RDR target transcripts by RNA-seq suggests a potentially significant functional overlap between RDR1, RDR2, RDR4, RDR5, and SGS3, while the rest of these proteins, RDR3, RDR6, and RDM12, do not display as much potential functional overlap. Similar GO terms among the different RDRs that we focused on in our study suggest that these RDRs may have redundant smRNA-generating substrates and thus biological functions. Previous studies demonstrated that over 99% of the annotated smRNA generating loci in Arabidopsis have a decrease of smRNAs in *rdr1/2/6* triple mutant plants, and there are 58 annotated *MIRNAs* that are RDR1/2/6-dependent [61]. By definition, miRNAs are derived from single-stranded hairpins in an RDR-independent manner. The finding that 58 annotated *MIRNAs* are downregulated in *rdr1/2/6* mutant plants suggests that RDR1, RDR2, and RDR6 may have redundant roles in regulating processes that were not previously uncovered by studying RDR1, RDR2, or RDR6 alone. Similarly, another study found that loss of RDR1/2/6 reduces the expression of antisense RNAs during plant response to abiotic stresses, and *TAS2* antisense RNAs were lower in *rdr1/6**, rdr2/6**,* or *rdr1/2/6* mutant plants when compared with *rdr6* single-mutant plants [62]. These findings reveal that RDR1/2/6 regulate the abundance of stress-related antisense RNAs cooperatively, but this functionality is separate from RDR6 functionality in siRNA biogenesis due to the fact that *TAS2* was still found to accumulate in the absence of RDR6 function (*rdr6* mutation). However, there is no study that currently addresses the redundancy of RDR3, RDR4, and RDR5 functions that is suggested in our study. Therefore, a global analysis of further *rdr* multimutants (e.g., *rdr3/4/5*) is likely to identify additional RDR-dependent smRNA-generating substrates and their specific targets in plants.

Furthermore, transcripts enriched for abiotic stress responses, such as water deprivation, salt, and ABA response, were found to be likely targets of RDR4, RDR5, RDR6, and SGS3. Since RDR6 and SGS3 are known to act cooperatively during siRNA biogenesis, we wanted to further test whether these two proteins also potentially function cooperatively to regulate specific plant abiotic stress responses by testing whether *rdr6* and *sgs3* mutant seedlings do not respond properly when subjected to osmotic and salt stress as well as ABA treatment. Indeed, we found that *rdr6* and *sgs3* mutants are both hypersensitive to ABA, tolerant to PEG8000, and less sensitive to salt treatment. Target prediction by psRNATarget revealed that two transcripts, *NCED3* and *MKK9*, may be the cause of the ABA hypersensitivity of *rdr6* and *sgs3* mutant seedlings. In support of this hypothesis, previous studies have demonstrated that *NCED3* overexpression results in plants that are more tolerant to drought than Col-0 [58], and that *mkk9* knockout mutants are hypersensitive to NaCl treatment [60]. These phenotypes match the expected phenotypes observed in *rdr6* and *sgs3* mutants, which should result in the overabundance of the mRNAs encoding both of these proteins. Future work will be needed need to explore this intriguing connection between RDR6 and SGS3-dependent smRNA regulation of *NCED3* and *MKK9* during plant ABA response as well as in response to salt and osmotic stress.

To find the origin of the smRNAs that target the *NCED3* and *MKK9* transcripts for potential PTGS-mediated silencing, we scanned for the smRNA-generating loci that demonstrated decreased levels of smRNAs that psRNATarget [39] predicted to bind the *NCED3* and *MKK9* RNAs in the *sgs3* and *rdr6* mutants. This analysis indicated that RDR6 and SGS3 use *cytochrome P450* transcripts to synthesize a 24 nt smRNA that can potentially bind to *NCED3* to perform gene silencing, along with two smRNAs (one 21 nt and one 22 nt), both processed from *TAS2* and have high levels of complementarity, to potentially target *MKK9* to regulate its transcript levels. Further analysis should be done to fully elucidate if RDR6- and SGS3-mediated silencing of *NCED3* and *MKK9* is the molecular mechanism influencing the ABA phenotypes of *rdr6* and *sgs3* mutant plants. 

Although previous studies have demonstrated that SGS3 works as an RNA stabilizer at the TAS loci during dsRNA formation mediated by RDR6 [26], the ability of this RBP to affect the levels of putative substrates for other RDR proteins has not been previously defined. Our findings would suggest that SGS3 likely functions with other RDRs in smRNA biogenesis. In our smRNA-seq data, differential smRNA abundance analysis from the six *rdr* mutants and *sgs3* revealed a larger population of smRNAs significantly decreased in *sgs3* as compared to *rdr6* mutants. Additionally, we identified significant overlaps of smRNA populations that are significantly decreased in *sgs3* and four other *rdr* mutants (*rdr1*, *rdr2*, *rdr4*, and *rdr5* mutants), in addition to the expected overlap with smRNAs lost in *rdr6* mutant plants. Furthermore, GO analysis of transcripts that show significant increases in abundance in RNA-seq datasets indicated that SGS3 regulates a broader range of transcripts than RDR6, including those encoding proteins involved in response to osmotic and cold stress. We also found overlap between transcripts misregulated in the absence of either SGS3 or RDR2, over-representing transcripts encoding proteins involved in regulation of the jasmonic acid pathway while the shared misregulated transcripts between SGS3 and RDR1 are over-represented for those encoding proteins involved in the response to wounding. Although further studies are needed, these findings lead to the intriguing hypothesis that SGS3 facilitates other RDRs in smRNA biogenesis, greatly expanding the functionality and substrates for this RBP. The strong connection between RDR- and SGS3-mediated smRNA-directed PTGS and numerous plant stress responses suggests this is an important area for future inquiry. In conclusion, our study provides a comprehensive transcriptome analysis unveiling new substrates and potential functions of the six Arabidopsis RDR proteins and their cofactors SGS3 and RDM12. Our findings also uncovered a new role for SGS3 in conjunction with RDR6 in abiotic stress response as well as new findings that SGS3 can also utilize putative substrates of other RDRs, suggesting it partners with these other proteins as well as RDR6.

## Figures and Tables

**Figure 1 ncrna-07-00028-f001:**
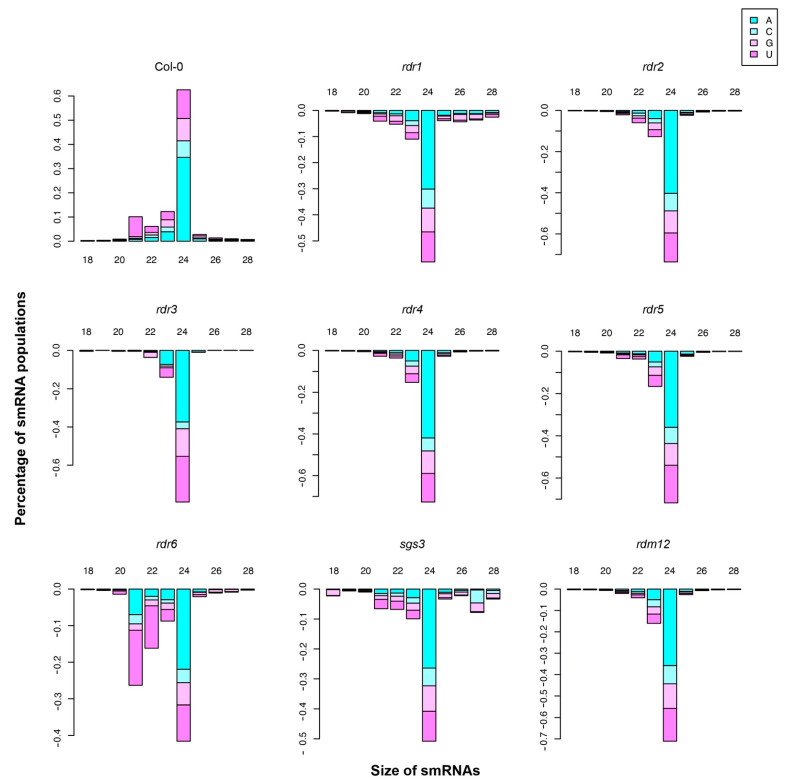
Small RNA population breakdowns for the six Arabidopsis RDR proteins and two of their cofactors (SGS3 and RDM12). The smRNAs were mapped to unbiased 500 nt genomic bins; the smRNA populations identified in Col-0 and smRNAs lost in *rdr1–6*, *rdm12*, and *sgs3* mutant plants (as specified) were plotted based on smRNA length and 5′ terminal nucleotide.

**Figure 2 ncrna-07-00028-f002:**
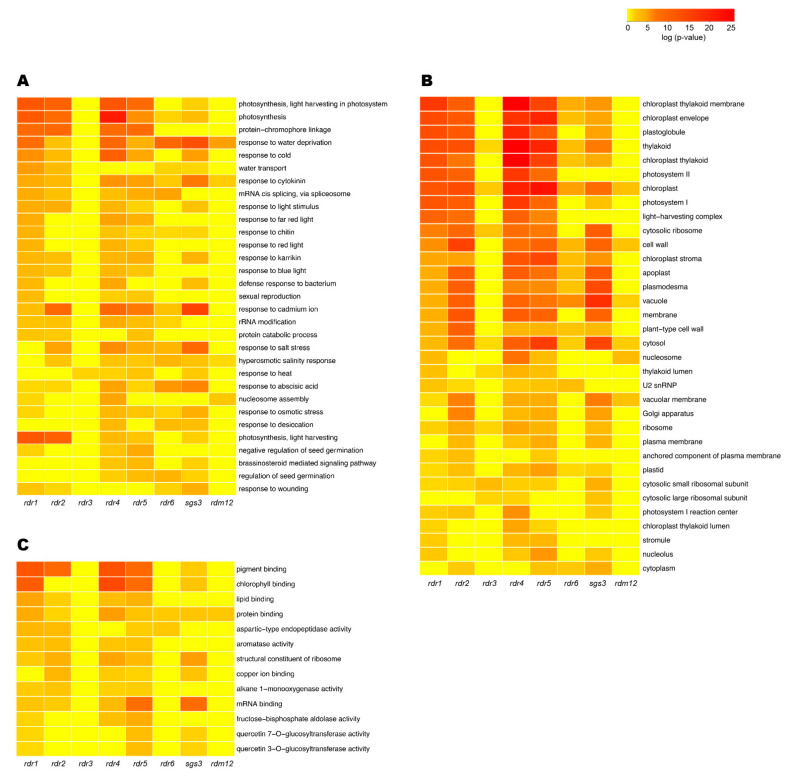
Heatmap of Gene Ontology (GO) enrichment terms for significantly more abundant transcripts in *rdr1–6*, *sgs3*, and *rdm12* mutants as compared to Col-0. The significantly more abundant transcripts (*p*-value < 0.05; edgeR) are plotted on the heatmaps and clustered by GO terms (*p*-value < 0.05; DAVID). Each column indicates a mutant genotype (as specified), each row presents a GO term (as specified), colors correspond to the log_10_ of *p*-value associated with each GO term, with red color showing the highest levels of significant enrichment within that GO category. (**A**) GO terms for biological processes. (**B**) GO terms for cell components. (**C**) GO terms for molecular function.

**Figure 3 ncrna-07-00028-f003:**
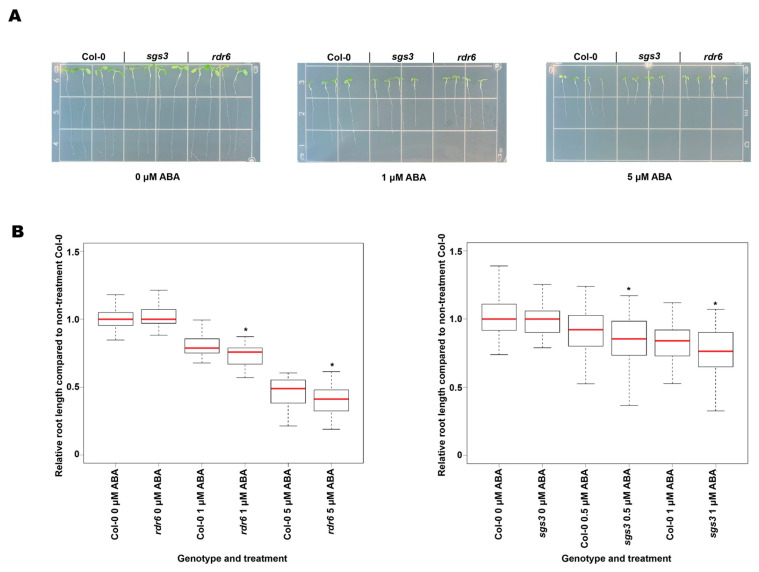
The *rdr6* and *sgs3* mutant plants are hypersensitive to ABA treatment. (**A**) Representative images of Col-0, *sgs3*, and *rdr6* seedlings grown under 0, 1, and 5 μM ABA treatment for five days. From left to right, the genotypes are as indicated. Seedlings were transferred to a new nontreatment plate for photographing. (**B**) (**Left**) Relative root length of Col-0 and *rdr6* seedlings under 0, 1, and 5 μM ABA treatment, *x*-axis denotes each genotype and ABA concentration used for treatment, *y*-axis shows the relative root length compared to Col-0 0 μM ABA treatment. A total of 185 seedlings were measured for these treatments. (**Right**) Relative root length of Col-0 and *sgs3* seedlings under 0, 0.5, and 1 μM ABA treatment. A total of 452 seedlings were measured for this treatment. * denontes *p*-values < 0.05; Mann-Whitney test.

**Figure 4 ncrna-07-00028-f004:**
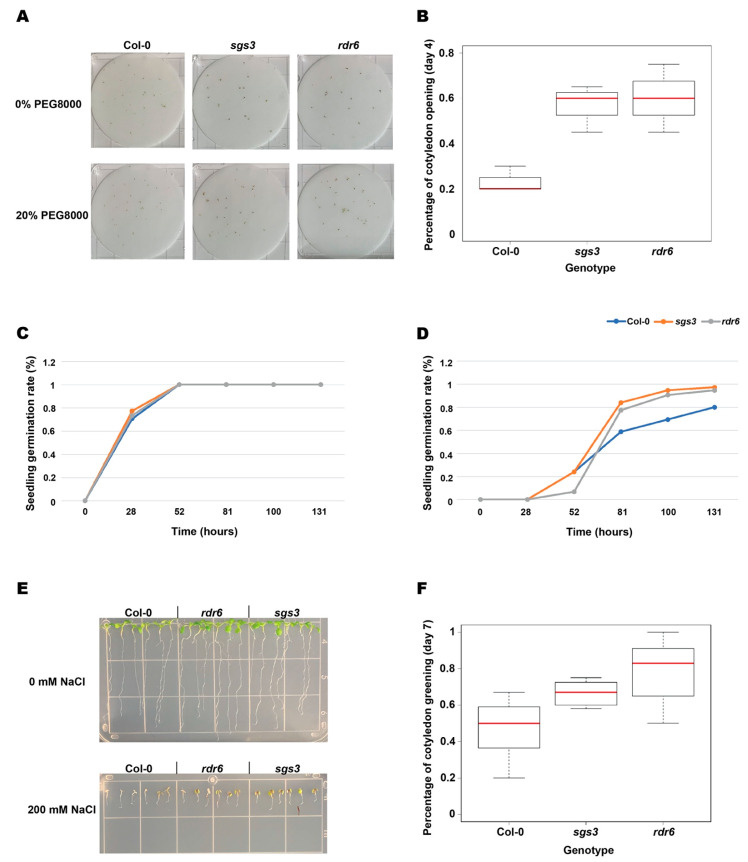
Results show *rdr6* and *sgs3* are less sensitive to PEG8000 and salt treatment. (**A**) Representative images of seedlings of the indicated genotypes grown without (**upper**) or with 20% PEG800 treatment in solution (**lower**) for four days. From left to right, the genotypes are Col-0, *sgs3*, and *rdr6*. (**B**) Percentage of cotyledon opening at day 4 with or without 20% PEG8000 treatment. The *x*-axis indicates the different genotypes, *y*-axis shows the percentage of cotyledon opening for each genotype. (**C**,**D**) The rate of seedling germination without (**C**) and with 200 mM NaCl (salt) (**D**) treatment. The *x*-axis of these graphs indicates developmental timing of the seedlings for the three genotypes, while the *y*-axis displays the percentage of germinated seedlings. Blue, orange, and grey lines denote Col-0, *sgs3*, and *rdr6*, respectively. (**E**) Respective images of seedlings of the indicated genotypes grown without (upper) or with 200 mM NaCl treatment (lower) for seven days. From left to right, the genotypes are Col-0, *rdr6*, and *sgs3*. (**F**) Percentage of cotyledon greening for seedlings of the three indicated genotypes after seven days of 200 mM NaCl treatment. The *x*-axis indicates the specific genotype, while the *y*-axis displays the percentage of seedlings with green cotyledons for each genotype.

**Table 1 ncrna-07-00028-t001:** Summary of putative substrates for the six RDRs and two cofactors identified in TAIR10 annotated transcripts. Two strand indicates putative substrates appeared on both the sense and antisense strands, and total means the total number of putative substrates identified. For annotated transcripts, strand information is given. The population of putative RDR substrates that was significantly downregulated in the mutants when compared with Col-0 are shown (smRNA counts down by 1/3, FDR < 0.05).

Genotype	Sense	Antisense	Two Strand	Total
*rdr1*	48	67	21	94
*rdr2*	12,488	2302	2261	12,529
*rdr3*	1	0	0	1
*rdr4*	37	50	24	63
*rdr5*	46	59	26	79
*rdr6*	74	104	54	124
*sgs3*	107	177	65	219
*rdm12*	136	183	72	247

**Table 2 ncrna-07-00028-t002:** Shared putative substrates among RDRs, RDM12, and SGS3. Shared substrates are counted by overlapping the union of sense and antisense differentially abundant putative substrates found in Table 1.

	*rdr1*	*rdr2*	*rdr3*	*rdr4*	*rdr5*	*rdr6*	*sgs3*
*rdr2*	10						
*rdr3*	1	1					
*rdr4*	7	10	1				
*rdr5*	5	13	1	8			
*rdr6*	3	9	0	3	4		
*sgs3*	3	69	0	6	8	27	
*rdm12*	7	58	1	9	11	5	14

## Data Availability

The raw and processed data for RNA-seq and smRNA-seq libraries made with RNA extracted from 30 days unopened flower buds of Col-0 and all mutants has been deposited in the NCBI Gene Expression Omnibus (http://www.ncbi.nlm.nih.gov/geo, accessed on 22 April 2021) database under the accession number GSE169270. The sequencing data presented here is also available through the EPIC-CoGe genome browser (Lyons and Freeling, 2008): https://genomevolution.org/coge/NotebookView.pl?nid=2906, accessed on 22 April 2021.

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
