# Peer review of "Global Analysis of RNA-Dependent RNA Polymerase-Dependent Small RNAs Reveals New Substrates and Functions for These Proteins and SGS3 in Arabidopsis"

_ncrna, 2021, doi:10.3390/ncrna7020028_

Round 1

Reviewer 1 Report

It was a delight to read a thorough work and well-written manuscript.
I do not have any major concerns with the soundness of the science presented here, the experiments and analysis are pretty straightforward, and the conclusions drawn are sound.  

I have one suggestion for the authors to consider adding one figure in the introduction that outlines the question asked and the experimental procedure. This type of 'graphical introduction' may be useful to readers that are not completely familiar with the subject.

Minor Comments

L148-152: There is some formatting error with different font style

L280-284: The same

Author Response

It was a delight to read a thorough work and well-written manuscript.
I do not have any major concerns with the soundness of the science presented here, the experiments and analysis are pretty straightforward, and the conclusions drawn are sound.  

I have one suggestion for the authors to consider adding one figure in the introduction that outlines the question asked and the experimental procedure. This type of 'graphical introduction' may be useful to readers that are not completely familiar with the subject.

We thank the Reviewer for these very kind comments on our work.

Minor Comments

L148-152: There is some formatting error with different font style

We have fixed the formatting as requested by the Reviewer.

L280-284: The same

We have fixed the formatting as requested by the Reviewer.

Reviewer 2 Report

In this manuscript, the authors investigated small RNA and mRNA expressions in the mutants of different RNA-dependent RNA Polymerases (RDR1-6) and two of their co-factors, SGS3 and RDM12. They provided a list of substrates and targets of these factors. They found that rdr1, 2, 4 and 5 affects on similar small RNA populations and mRNAs in similar biological pathways. Moreover they found that SGS3 may acts as general co-factor of most RDRs. They also found that rdr6 and sgs3 up regulates genes related to abiotic stresses and showed that these mutants are required for proper response to abiotic stresses.

I think this study provides useful information to the plant research communities of both small RNAs and abiotic stresses, and good for publication in this journal. 

I just have a few comments:

1) On of the limitations of this study is that they probably miss substrates and targets of RDRs if two or more RDRs act redundantly on a same substrate. For example, because rdr1, 2, 4 and 5 affect mRNA expressions in similar biological pathways according to the GO term enrichment analysis shown in Figure 2, it is likely that these RDRs share substrates. So I think if they use double/triple/quadruple mutants of these RDRs, they may be able to examine their substrate redundancies and identify more RDR substrates. I believe the authors should provide more discussions for how the functional redundancies among RDRs can affect their substrate/target identification.

2) Line 416-439: No detailed data is provided for these descriptions. How much genes were found to be dependent on both RDR6 and SGS3? What was the criteria to determine the expression a gene is dependent on them? Is it possible to give scores for the dependency? Are NCED3 and MKK9 high ranker according to such score?  

3) Figure 3B: Why not combining the left and the right panels of the box plots?

4) Figure 4: the legends for C and E are swapped.

Author Response

In this manuscript, the authors investigated small RNA and mRNA expressions in the mutants of different RNA-dependent RNA Polymerases (RDR1-6) and two of their co-factors, SGS3 and RDM12. They provided a list of substrates and targets of these factors. They found that rdr1, 2, 4 and 5 affects on similar small RNA populations and mRNAs in similar biological pathways. Moreover they found that SGS3 may acts as general co-factor of most RDRs. They also found that rdr6 and sgs3 up regulates genes related to abiotic stresses and showed that these mutants are required for proper response to abiotic stresses.

I think this study provides useful information to the plant research communities of both small RNAs and abiotic stresses, and good for publication in this journal. 

We thank the Reviewer for these very kind comments on our work.

I just have a few comments:

1) On of the limitations of this study is that they probably miss substrates and targets of RDRs if two or more RDRs act redundantly on a same substrate. For example, because rdr1, 2, 4 and 5 affect mRNA expressions in similar biological pathways according to the GO term enrichment analysis shown in Figure 2, it is likely that these RDRs share substrates. So I think if they use double/triple/quadruple mutants of these RDRs, they may be able to examine their substrate redundancies and identify more RDR substrates. I believe the authors should provide more discussions for how the functional redundancies among RDRs can affect their substrate/target identification.

To address this concern, we have added the following discussion of this shortcoming to the Discussion section of our manuscript.

L478-497: Similar GO terms among the different RDRs that we focused on in our study suggests that these RDRs may have redundant smRNA-generating substrates, and thus biological functions. Previous studies demonstrated that over 99% of the annotated smRNA generating loci in Arabidopsis have a decrease of smRNAs in rdr1/2/6 triple mutant plants, and there are 58 annotated MIRNAs that are RDR1/2/6dependent [57]. MiRNAs, by definition, are derived from singlestranded hairpins in an RDRindependent manner. The finding that 58 annotated MIRNAs are downregulated in rdr1/2/6 mutant plants suggests that RDR1, 2, and 6 may have redundant roles in regulating processes that were not previously uncovered by studying RDR1, 2, or 6 alone. Similarly, another study found that loss of RDR1/2/6 reduces the expression of anti-sense RNAs during plant response to abiotic stresses, and TAS2 antisense RNAs were lower in rdr1/6, rdr2/6, or rdr1/2/6 mutant when compared with rdr6 single mutant plants [58]. These findings reveal that RDR1/2/6 regulate the abundance of stress related anti-sense RNAs cooperatively, but this functionality is separate from RDR6 functionality in siRNA biogenesis due to the fact that TAS2 was still found to accumulate in the absence of RDR6 function (rdr6 mutation). However, there is no study that currently addresses the redundancy of RDR3, 4, and 5 function that is suggested in our study. Therefore, a global analysis of further rdr multi-mutants (e.g. rdr3/4/5) is likely to identify additional RDR-dependent smRNA-generating substrates and their specific targets in plants.”

L827-831: “[57] Polydore, S.; Axtell, MJ. Analysis of RDR1/RDR2/RDR6independent small RNAs in Arabidopsis thaliana improves MIRNA annotations and reveals unexplained types of short interfering RNA loci. Plant J. 2018, 94(6), 1051-1063.

[58] Matsui, A.; Iida, K.; Tanaka, M.; Yamaguchi, K.; Mizuhashi, K.; Kim, JM.; Takahashi, S.; Kobayashi, N.; Shigenobu, S.; Shinozaki, K.; Seki, M. Novel Stress-Inducible Antisense RNAs of Protein-Coding Loci Are Synthesized by RNA-Dependent RNA Polymerase. Plant Physiol. 2017, 175(1), 457-472.”

References [57]-[61] have been changed to [59]-[63].

2) Line 416-439: No detailed data is provided for these descriptions. How much genes were found to be dependent on both RDR6 and SGS3? What was the criteria to determine the expression a gene is dependent on them? Is it possible to give scores for the dependency? Are NCED3 and MKK9 high ranker according to such score?  

We have added this information to the manuscript and the specific changes are pasted below. We have also added a Supplemental Table 3 to address this comment which describes all of the smRNAs that are significantly less abundant in rdr6and sgs3 mutant as compared to Col-0 plants.

L432: “(p-value < 0.01; EdgeR)

L433-434: “(Supplemental Table 3)”

L435-436: “(p-value < 0.05; Edge R)

L437: added “86 and 100”, deleted “several”

L438: added “, respectively,”

L439: added “16

L440: added “shared by RDR6 and SGS3

3) Figure 3B: Why not combining the left and the right panels of the box plots?

Unfortunately, these figures do not easily combine given that slightly different concentrations of ABA were used for the treatments of the rdr6 and sgs3 mutant plants. Additionally, we feel that having separate figures for each mutant line makes these figures very simple and straight-forward for the readers of the manuscript. Due to these reasons, we have not combined these figures in the revised version of the manuscript.

4) Figure 4: the legends for C and E are swapped.

We have fixed the Figure 4 legend as directed.

Reviewer 3 Report

The aim of this work is to explore the function of the six RDRs present in Arabidopsis as well as two accessory proteins in the generation of smRNAs. For this purpose they perform RNASeq of small RNAs in wild type and null mutants of each RDR. The analysis of the data allows the identification of substrates for each RDR. In addition they perform a regular RNASeq, that combined with informatic analysis helps to identify targets of smRNAs. All the data provided represent an invaluable repository for researchers interested in the function of RDRs and smRNAs.

They identify RDR6 and SGS3 as responsible of generating sRNAs against transcripts related to abiotic stress, salt stress and ABA response. By careful phenotypic analysis they confirm that rdr6 and sgs3 mutants are hypersensitive to ABA, more resistant to osmotic stress and more tolerant to salt.

The paper is well written, the results are clearly explained. I find this work a relevant contribution to the field of ncRNAs.

Author Response

The aim of this work is to explore the function of the six RDRs present in Arabidopsis as well as two accessory proteins in the generation of smRNAs. For this purpose they perform RNASeq of small RNAs in wild type and null mutants of each RDR. The analysis of the data allows the identification of substrates for each RDR. In addition they perform a regular RNASeq, that combined with informatic analysis helps to identify targets of smRNAs. All the data provided represent an invaluable repository for researchers interested in the function of RDRs and smRNAs.

They identify RDR6 and SGS3 as responsible of generating sRNAs against transcripts related to abiotic stress, salt stress and ABA response. By careful phenotypic analysis they confirm that rdr6 and sgs3 mutants are hypersensitive to ABA, more resistant to osmotic stress and more tolerant to salt.

The paper is well written, the results are clearly explained. I find this work a relevant contribution to the field of ncRNAs.

We thank the Reviewer for these very kind comments on our work. We agree that this will be an important data repository for the plant small RNA research community.